# *Bacillus paralicheniformis* RP01 Enhances the Expression of Growth-Related Genes in Cotton and Promotes Plant Growth by Altering Microbiota inside and outside the Root

**DOI:** 10.3390/ijms24087227

**Published:** 2023-04-13

**Authors:** Jinzhi Xu, Lijun Qin, Xinyi Xu, Hong Shen, Xingyong Yang

**Affiliations:** 1College of Pharmacy, Chengdu University, Chengdu 610052, China; xjz_swu@126.com (J.X.); qlj0803@163.com (L.Q.); xuxinyi0669@163.com (X.X.); 2Antibiotics Research and Re-Evaluation Key Laboratory of Sichuan Province, Chengdu University, Chengdu 610052, China; 3College of Resources and Environment, Southwest University, Chongqing 400715, China

**Keywords:** plant growth-promoting bacteria, *Bacillus paralicheniformis* RP01 strain, root length, plant growth, growth-related genes

## Abstract

Plant growth-promoting bacteria (PGPB) can promote plant growth in various ways, allowing PGPB to replace chemical fertilizers to avoid environmental pollution. PGPB is also used for bioremediation and in plant pathogen control. The isolation and evaluation of PGPB are essential not only for practical applications, but also for basic research. Currently, the known PGPB strains are limited, and their functions are not fully understood. Therefore, the growth-promoting mechanism needs to be further explored and improved. The *Bacillus paralicheniformis* RP01 strain with beneficial growth-promoting activity was screened from the root surface of *Brassica chinensis* using a phosphate-solubilizing medium. RP01 inoculation significantly increased plant root length and brassinosteroid content and upregulated the expression of growth-related genes. Simultaneously, it increased the number of beneficial bacteria that promoted plant growth and reduced the number of detrimental bacteria. The genome annotation findings also revealed that RP01 possesses a variety of growth-promoting mechanisms and a tremendous growth-promoting potential. This study isolated a highly potential PGPB and elucidated its possible direct and indirect growth-promoting mechanisms. Our study results will help enrich the PGPB library and provide a reference for plant–microbe interactions.

## 1. Introduction

Plant growth-promoting bacteria (PGPB) are bacteria found around plant roots (rhizosphere) that can promote plant growth. PGPB can promote plant growth directly by: (a) fixing nitrogen, solubilizing phosphate and releasing potassium; (b) secreting/inducing the production and release of phytohormones; and (c) producing siderophores. Additionally, they can indirectly promote plant growth by: (a) secreting antibiotics to control pathogens; (b) improving plant resistance to abiotic stresses (drought and salinity); and (c) improving the rhizosphere environment (recruitment of beneficial microorganisms) [1,2]. For example, four *Streptomyces* strains (HM2, HM3, HM8 and HM10) have been reported to enhance cucumber growth and yield through various mechanisms including indole acetic acid (IAA) production, siderophore excretion and phosphate solubilization [3]. *Sphingomonas* sp. Hbc-6 recruits beneficial rhizosphere bacteria to increase maize biomass and drought tolerance [4]. These attributes enable PGPB to supplant synthetic chemical fertilizers, thereby preventing environmental pollution and further bioremediating and controlling plant pathogens [5].

PGPB isolation and evaluation are critical not only for practical applications, but also for basic research. Currently, there are fewer known PGPB strains with comprehensive functions, and their growth-promoting mechanisms need further improvement and exploration. Hence, PGPB, with excellent growth-promoting activity on various plants while having the ability to adapt to various environmental conditions, must be investigated thoroughly. The *Bacillus paralicheniformis* RP01 strain, having a good pro-growth effect on various plants (cabbage, tobacco and cotton), was screened from the root surface of *Brassica chinensis* using a phosphate-solubilizing medium (Appendix A). Recent studies have found that some *B. paralicheniformis* strains can promote plant growth and antagonize pathogens [6,7]. The *B. paralicheniformis* TRQ65 strain was isolated from the rhizosphere of durum wheat (*Triticum turgidum* subsp. *durum*) [8]. The individual inoculation of TRQ65 in wheat seedlings showed significant increases in biomass. Further, it inhibited mycelial growth of the wheat phytopathogen *Bipolaris sorokiniana*, the causal agent of spot blotch [8,9]. *B. paralicheniformis* MDJK30, isolated from the rhizosphere of the peony in Shandong, China, inhibited the fungi *Fusarium solani*, which can cause root rot [10]. We investigated the growth-promoting mechanism of *B. paralicheniformis* RP01 using upland cotton (*Gossypium hirsutum*) Yumian-1 as the host to increase crop yield and improve the soil conditions for sustainable agriculture. This study may provide added fundamental knowledge to the PGPB library and serve as a reference for plant–microbe interactions.

## 2. Results

### 2.1. RP01 Isolation and Its Physiological and Biochemical Characterization

The RP01 strain with P-solubilizing ability was isolated from purple soil from the root surface of *Brassica chinensis* L. using a P-solubilizing Pikovskaya (PKO) agar medium. Compared to the control, the soluble phosphorus content in the fermentation broth inoculated with RP01 was about 129 ± 11.89 mg/L, which was the highest among the strains isolated in the same batch (Appendix A). The colonies were white, irregular and had slightly convex edges with a moist, transparent gel on the surface (Figure 1a). The colonies were 2–3 mm in diameter at 1–2 d of incubation and 5–6 mm in diameter at 3–5 d of incubation. The bacterium was rod-shaped and approximately 2–3 µm long, and its surface was uneven (Figure 1b).

RP01 could use citrate and xylose as carbon sources, but not hydrolyze starch, and it could ferment glucose to produce acid and acetylmethylmethanol, but not gas. Furthermore, RP01 was able to liquefy gelatin and hydrolyze hydrogen peroxide (Appendix A).

Sequencing of 16S rRNA and evolutionary tree analysis revealed that RP01 belongs to *Bacillus paralicheniformis* (Figure 1c). The complete taxonomic description was as follows: *Firmicutes*, *Bacilli*, *Bacillales*, *Bacillaceae*, *Bacillus* and *B. paralicheniformis* RP01.

### 2.2. Differences in Hosts after Inoculation with RP01

Our previous tests on various plants showed that inoculation with RP01 had a strong growth-promoting effect (Appendix A). The results of cotton inoculation showed that the biomass (plant height, root length, leaf width, etc.) of cotton seedlings inoculated with 10^4^ cfu/mL and 10^8^ cfu/mL RP01 was considerable higher than that of the biomass in sterile water control after 30 d of growth and did not differ in the concentration gradient; additionally, the growth-promoting effect was the same for both low and high inoculations (Figure 2a). Therefore, we compared cotton seedlings inoculated with low concentrations (10^4^ cfu/mL, inoculation group) of RP01 and sterile water control (mock group) to analyze the promotion of plant growth.

PGPB often regulates growth by modulating plant hormones [11]; therefore, we examined hormone levels in plant leaves with an enzyme-linked immunosorbent assay (ELISA). It was found that brassinosteroids (BRs), which have a pro-growth effect on plants, considerably increased (43.9%) after inoculation, and auxin (indole acetic acid, IAA) was slightly higher (5.1% increase) than the control (Figure 2b). Jasmonic acid (JA) and salicylic acid (SA), which regulate plant defense responses, followed the same trend as IAA, increasing by 5.5% and 7.5%, respectively (Figure 2c).

The response of cotton roots was more critical because *B. paralicheniformis* RP01 was directly inoculated in the rhizosphere. We used root tissues for fluorescence quantification and found that the IAA biosynthesis gene *YUCCA4* (*p* = 0.0019) [12], signal transduction gene *AUX1* (*p* = 0.0005) [13], BR positive regulator *BES1* (*p* = 0.0004) [14], and gibberellic acid (GA) biosynthesis genes *GA20ox* (*p* = 0.0054) and *GA3ox* (*p* = 0.0004) [15] were significantly upregulated (Figure 2d). Meanwhile, disease-related genes such as JA biosynthesis genes *AOC1* (*p* < 0.0001) [16] and *OPR3* (*p* = 0.0005) [17], the JA biosynthesis gene *TCP* (*p* = 0.0396) [18], JA response genes *PDF1.2* (*p* = 0.0012) [19] and ERF1 (*p* < 0.0001) [20], and SA biosynthesis-related genes *ICS1* (*p* = 0.0003) [21] and *EDS1* (*p* < 0.0001) [22] were significantly upregulated. However, the expression of the SA biosynthesis-related gene *PAD4* [22] remained unchanged (Figure 2e).

### 2.3. Microbial Diversity inside and outside the Cotton Roots after Inoculation with RP01

The microbial composition inside and outside the roots is critical for plant growth and development [23,24,25]. Thirty days after inoculation, rhizosphere soil samples and cotton seedling root samples from each group were extracted to assess microbial communities based on 16S rDNA amplicon sequencing. Species richness and community diversity (alpha-diversity) both increased after inoculation, but were not significantly different (Appendix A).The species composition at the phylum level is shown in Figure 3a: after inoculation with RP01, *Proteobacteria* increased from 42.85 to 49.63% outside the root, but decreased from 95.02 to 81.44% inside the root; *Bacteroidota* decreased from 21.86 to 13.26% outside the root, but increased from 1.04 to 3.29% inside the root; and *Actinobacteriota* increased both inside and outside the roots. The top three dominant genera in the rhizosphere were *WCHB1-32*, *Streptomyces*, and *Bradyrhizobium* (Figure 3b), while in the roots they were *unclassified*_*Rhodocyclaceae*, *Hydrogenophaga*, and *unclassified*_*Comamonadaceae* (Figure 3c).

Appendix A visualizes the top 10 genera in abundance. In the rhizosphere, the abundance was higher for *Candidatus_Solibacter* (*p* = 0.041), *Gemmatimonas* (*p* = 0.015), *Opitutus* (*p* = 0.045) and *Sphingomonas* (*p* = 0.008). The first two significantly increased after inoculation with RP01, while the last two significantly decreased after inoculation (Figure 4a). Within the roots, only the abundance of *unclassified*_*Rhodocyclaceae* (*p* = 0.038) was dominant, which had decreased significantly after inoculation with RP01 (Figure 4b). As seen in the network association diagram of genera to genera [26], the top 20 genera of inter-roots were more simply related than those of intra-roots (Figure 4c,d). In the rhizosphere, the most represented genus *WCHB1-32* negatively correlated with *Caulobacter* and had a weaker positive correlation with *Lentimicrobium*. The previously screened differential genus *Candidatus*_*Solibacter* positively correlated with *Gemmatimonas*, whereas three other genera negatively correlated with *Gemmatimonas*. The top dominant genus, *unclassified*_*Rhodocyclaceae*, was negatively correlated within the roots, except for with *unclassified*_*Burkholderiales*. We performed the PICRUSt2 functional prediction [27] of the microbiome in the rhizosphere and found that at the enzyme level, histidine kinase (2.7.13.3) and 3-oxoacyl-[acyl-carrier-protein] reductase (1.1.1.100) were upregulated after inoculation with RP01 (Appendix A). PICRUSt2 functional predictions within the roots indicated that, at the enzyme level, NADH: ubiquinone reductase (H(+)-translocating) (1.6.5.3), DNA-directed DNA polymerase (2.7.7.7), DNA helicase (3.6. 4.12), histidine kinase (2.7.13.3) and peptidylprolyl isomerase (5.2.1.8) were downregulated (Appendix A).

We also sequenced ITS amplicons of the roots to analyze the changes in fungal diversity changes. We found no significant difference in fungal species, while the community diversity was higher than that of the control (Appendix A; Appendix A). Although more than 95% of the dominant genera were *unclassified*_*fungi* and *unclassified*_*Ascomycota,* the relative percentage of *unclassified*_*Ascomycota* decreased after inoculation (Appendix A).

### 2.4. Genome and Comparative Genome of RP01

To further analyze the growth-promoting mechanism of RP01, we analyzed the RP01 genome in detail. RP01 contains only one chromosome with a total genome length of 4,338,611 bp and a GC content of 45.95%. The summarized results from the six major databases (NR, Swiss-Prot, Pfam, EggNOG, GO and KEGG) are listed in Appendix A. Finally, a closed-loop circle map was obtained using Circos analysis [28] to fully characterize the genome (Figure 5).

The RP01 genome contains 159 CAZymes [29] (Figure 6a and Appendix A), including genes encoding cellulase, chitinase, α- and β-glucosidase, α-amylase, and xylosidase, indicating its ability to utilize multiple carbon sources. Simultaneously, a large number of genes involved in the glycolysis, pentose phosphate and tricarboxylic acid cycles were predicted. The cation transporter, H^+^-transporting ATPase (F-ATPase) and Na^+^/H^+^ transporters were also encoded in the RP01 genome. Among them, two K^+^ transporter systems, the Kdp and Trk systems, are present in RP01. Moreover, we also identified Na^+^/H^+^ reverse transporters (*Cpa1*, *Cpa2* and *NhaC*). In terms of motility, RP01 possesses genes related to flagellar biosynthesis and assembly, such as *fliDEGJMSTPW*, *flhABF* and *flgBCGL*, and a carbon storage regulator gene (*csrA*). In addition, genes involved in chemotaxis, such as *cheABCDRWY*, are protein-coding genes of a two-component system of methyl-accepting chemotaxis. The two-component system is helpful for signal recognition of exudates and adaptation to the environment. The presence of these genes indicates that RP01 can respond to stimuli and move toward the plant roots.

Secondary metabolite analysis of the RP01 genome using anti-SMASH [30] predicted 12 clusters of secondary metabolite genes (Appendix A), including NRPS, bacteriocins, lanthipeptides, terpenes, lasso peptides and siderophores. Metabolites with diverse biological activities are mostly antibacterial in nature. Five of these were compared to the database: lichenysin, butirosin A/butirosin B, fengycin, bacitracin and bacillibactin (Figure 6b). Based on the CARD database annotation [31], the RP01 genome contains a total of 209 drug resistance genes (Figure 6c), which mainly includes 29 types of macrolide antibiotics, polypeptide antibiotics, glycopeptide antibiotics, aminoglycoside antibiotics, etc. (Appendix A). The RP01 genome also has protein quality control system-related genes (*htpG*, *htpX*, *dnaK*, *dnaJ* and *groEL*) and genes related to adaptation for survival in extreme environments, such as genes involved in glycine betaine synthesis and transport (*betB*, *proVWX* and *opuAD*) and the trehalose operon repressor gene. RP01 encodes numerous genes related to heavy metal transport and detoxification, such as various heavy metal transport proteins, including the magnesium transporter protein (*mgtE*), zinc transporter protein and arsenite efflux transporter protein (*arsB*), which transport metal ions from the cytoplasm to the outside of the cell. The *arsR* gene encoding the ArsR/SmtB family of trans-acting blocking proteins and arsenate reductase (*arsC*) was also identified in the genome.

The RP01 genome also contains alkaline phosphatase and phosphate transporters (*phoADE* and *pstABCS*), nitrogen assimilation genes and their regulatory elements (*glnABKLRT*), nitrate reduction and transporter clusters (*narHGKI* and *nasABCDEF*), the urease gene cluster (*ureABCDEFG*), sulfate reduction-related enzymes (*cysCHJI*), and ionophores (*IucA* and *IucC*) that promote plant growth and development. L-tryptophan (L-TRP) is a critical residue required for normal plant growth and development, and it acts as a precursor for plant growth regulators [32,33]. We identified the complete biosynthetic pathway (*trpABCDEF*) of L-TRP, agmatinase (*speB*) and spermine synthase (*speE*), which can catalyze the conversion of amino acids into plant growth-promoting compounds [34].

In addition, we selected three *B. paralicheniformis* strains with different functions for comparative genomic analysis with RP01. The basic annotation information is presented in Table 1. Among these, strain BIK4, which was isolated from the rice rhizosphere, could promote rice growth and inhibit pathogens [35]. KJ-16^T^ was isolated from a soybean fermentation paste [36]. ES-1 was isolated from saline-sodic soil and showed salt tolerance and antibacterial activity against various pathogens [37]. From the phylogenetic tree of housekeeping genes, it was found that RP01 had the highest similarity with BIK4 (Figure 7a), and BIK4 happens to be the rhizosphere growth-promoting bacteria. Based on the Venn diagram (Figure 7b), we found that 3750 genes were shared by the 4 bacteria with KEGG functional enrichment, as shown in Figure 7c (Appendix A), and metabolic pathways were significantly enriched. RP01 was found to contain 91 unique genes, including the phosphatase RapE regulator, thioredoxin reductase, Fe-S cluster biogenesis protein NfuA and tellurite resistance membrane protein TerC (Appendix A).

## 3. Discussion

### 3.1. Plant Growth-Promoting Function of RP01

Through the assessment of RP01, it was found that RP01 has strong viability: it can decompose a variety of macromolecular carbons, has a variety of respiratory modes to obtain energy, has broad-spectrum drug resistance, can secrete antibacterial substances, and can adapt to extreme environments and a variety of stress strategies. For example, the RP01 genome contains a fengycin gene cluster. Fengycin has broad-spectrum antibacterial activity, low toxicity and low drug resistance [38]. Carbon storage regulator A (*csrA*) is an RNA-binding protein that plays an important regulatory role in various physiological processes such as bacterial carbon metabolism [39], biofilm formation [40], motility [41], quorum sensing [42] and stress response [43]. In addition, the motility and chemotaxis of RP01 contribute to its colonization of the rhizosphere of plants, which plays a vital role in promoting plant growth.

RP01 has a phosphate-solubilizing function (*phoADE*) that can convert inorganic phosphorus/organic phosphorus into phosphorus that can be directly absorbed and utilized by plants. At the same time, RP01 has a nitrate transporter (*narHGKI*) and nitrate reductase (*nasABCDEF*) involved in nitrate transport and reduction, which were previously found in the plant growth-promoting *Bacillus subtilis* MBI 600 [44]. Additionally, a urease (*ureABCDEFG*) gene cluster in the RP01 genome can hydrolyze urea into ammonia. Subsequently, glutamine synthetase (*glnA*) converts NH_4_^+^ produced during nitrate assimilation into glutamine; at the same time, glutamate dehydrogenase (*gdhA*) can also produce NH_4_^+^ to produce glutamate. Glutamine and glutamic acid are the sources of amino acids in biosynthesis that ultimately participate in synthesizing microbial nucleic acids and proteins. Thus, RP01 has an ammonium assimilation enzyme system with glutamine synthetase/glutamate synthase (GS/GOGAT) and glutamate dehydrogenase (GDH) activities to regulate nitrogen absorption and transformation, thereby providing nitrogen for plant growth. RP01 also contains sulfate reduction and transport-related genes (*cysCHJI*) that promote plant growth and seed germination [45].

The genome of strain RP01 contains the spermidine synthase-related gene (*speAGE*), which is related to spermine biosynthesis. Spermidine is essential for plant cell viability and is associated with lateral root expansion, plant pathogen resistance and alleviating osmotic, oxidative and acidic stress [34]. We identified the complete synthesis pathway of L-TRP (*trpABCDEF*). Studies have found that spraying L-TRP onto maize leaves can promote plant growth [46]. PGPB also synthesizes chitinase, a secondary metabolite that protects plants against disease. RP01 encodes chitinase to hydrolyze the cell wall of disease-causing fungi to help plants protect against diseases. In addition, the genes *gabD* and *gabR*, which are involved in synthesizing γ-aminobutyric acid for disease/pest suppression [34], were identified. However, *opuC*, *opuA*, *proX*, *proV* and *proW* may also protect plants from oxidative stress [33].

### 3.2. RP01 Affects Inter-Root and Intra-Root Microorganisms

*Bacillus* contains a variety of antagonistic bacteria that can be widely used in the prevention and treatment of plant pathogens. Competition is a common antagonistic mechanism of *Bacillus*. RP01 has broad-spectrum resistance and bacteriostatic effects that can affect the growth of rhizosphere microorganisms. Therefore, RP01 competes with the original rhizosphere microorganisms after inoculation, affecting the composition and changes in microorganisms and regulating plant growth.

*Actinobacteriota* in the rhizosphere and roots increased significantly after inoculation with RP01 (Figure 3a). Crop root diseases are greatly reduced or eliminated after inoculating *Actinobacteriota* into the soil. In addition, it can also significantly increase the crop root, stem and leaf biomass yield and enhance crop disease resistance [47,48]. *Streptomyces* produces 90% of the antibiotics of *Actinobacteriota.* It produces antibiotics and enzymes related to carbohydrate hydrolysis, which can inhibit the growth of pathogenic bacteria [49]. *Streptomyces albidoflavus* St-220 can promote the growth of alfalfa by solubilizing phosphorus, fixing nitrogen, producing auxin and antagonizing *Rhizoctonia solani* [50]. *Streptomyces hygroscopicus* OsiSh-2 can improve resistance against rice blast pathogens and enhance chloroplast development to promote rice growth [51]. The results showed that *Streptomyces* increased significantly in the rhizosphere after inoculation with RP01 (Figure 3b). *Bradyrhizobium*, which can expand and form nodules in the roots of host plants and fix atmospheric nitrogen to bound nitrogen (ammonia) for use by host plants, was also significantly upregulated after inoculation (Figure 3b). In the *Atractylodes lancea*–maize intercropping system, intercropping can promote the enrichment of plant growth-promoting bacteria, including *Streptomyces*, *Bradyrhizobium*, *Candidatus*_*Solibacter* and *Gemmatirosa*, thereby promoting the growth of *A. lancea* [52]. After inoculation with RP01, *Candidatus*_*Solibacter* and *Gemmatirosa* increased significantly in the rhizosphere, indicating that RP01 contributed to the enrichment of growth-promoting bacteria. (Figure 4a). In addition, *Acinetobacter* and *Actinoplanes* in the roots increased after inoculation with RP01 (Figure 3c). *Acinetobacter* sp. SuKIC24 promotes plant growth through phosphorus solubilization and IAA production [53]. *Actinoplanes* can produce IAA, indole-3-pyruvic acid (IPyA) and GA3, increase plant disease resistance, and promote plant growth [54]. In summary, some plant growth-promoting bacteria were increased and detrimental bacteria were inhibited after inoculation with RP01.

### 3.3. Interaction of RP01 with Cotton

After 30 d of inoculation with RP01, the biomass indicators of cotton significantly increased (Figure 2a). The same pro-growth effect was observed at high and low concentrations, indicating that a low concentration (10^4^ cfu/mL) achieved an excellent growth-promoting effect. The levels of BR and IAA, which promoted plant growth, were increased after inoculation, and BR had the most significant increase of 43.9% (Figure 2b). Similarly, root hormone-related genes were also upregulated. Interestingly, the root length of RP01 cotton increased most significantly after inoculation (Appendix A and Figure 2), and the significant increase in plant root length was more conducive to nutrient acquisition. The levels of the hormones SA and JA, which regulate plant defense responses, also increased after inoculation with RP01 (Figure 2c), but not significantly. Together with the increase in rhizosphere microorganisms after inoculation (Figure 3), the activation of SA and JA pathways in cotton can also be elucidated. However, this is because of the limited number of pathogens and the protective mechanisms of probiotics (*Actinoplanes*, *Bradyrhizobium* and *Acinetobacter*) and RP01. Therefore, the responses of the plant hormones SA and JA were not obvious.

We described the possible direct and indirect growth-promoting mechanisms of RP01 by analyzing the genome and microbiome. In summary, RP01 had a growth-promoting effect on various crops (Appendix A). The genome also showed that RP01 has various growth-promoting methods and tremendous growth-promoting potential. Simultaneously, RP01 can also produce fengycin and other secondary metabolites with unknown functions, which may play a role in agricultural and pharmaceutical applications.

## 4. Materials and Methods

### 4.1. Bacterial Strain

According to the method of Shen et al. [55], P-solubilizing Pikovskaya (PKO) agar medium was used to isolate RP01 from the root surface of *Brassica chinensis* L. grown in the Beibei District (30°26′12″ N, 106°26′25″ E), Chongqing, China. Bacterial suspensions were prepared by culturing cells in 50 mL of LB medium (10 g/L of tryptone, 5 g/L of yeast extract and 10 g/L of NaCl) in 300 mL flasks on a rotary shaker (150 rpm) at 37 °C for 12 h. Scanning electron microscopy was used for morphological observations [56]. The phenotypical, physiological and biochemical characteristics of RP01 were determined using the methods described by Samina et al. [57]. These characteristics included gelatin liquefaction, glucose-produced acid and gas, xylose-produced acid, the methyl red test, the presence of catalase and oxidase, phenylalaninase, starch hydrolysis, the use of citrate, and the Voges–Proskauer test (Appendix A).

### 4.2. Molecular Identification

After culturing RP01 in LB medium for 48 h (37 °C, 150 rpm), bacterial cells were collected by centrifugation (4500× *g* for 5 min at 4 °C). For preliminary confirmation of the isolated bacterial strain, a polymerase chain reaction (94 °C for 10 min, followed by 34 cycles at 94 °C for 30 s, 56 °C for 30 s and 72 °C for 90 s, with a final extension at 72 °C for 10 min) was performed with universal 16S primers 27F (5′-AGAGTTTGATCCTGGCTCAG-3′) and 1492R (5′-TACGGTTACCTTGTTACGACTT-3′). The PCR products were sequenced by BGI (Shenzhen, China), and the resulting sequences were compared with those of 20 strains in the GenBank database using MEGA 11.0 [58], based on the neighbor-joining method.

### 4.3. Evaluation of Growth Promotion in Cotton Seedlings

The bacteria were resuspended and diluted in deionized water (0 cfu/mL for the mock group, and 10^4^ cfu/mL and 10^8^ cfu/mL for the test group). Upland cotton (*G. hirsutum*) Yumian-1 seeds were subjected to surface sterilization with 20% (*v*/*v*) H_2_O_2_ for 2 min, and then were individually sown in plastic pots (12 cm diameter and 16 cm height). The soil was collected at a depth of 0–15 cm from the campus (30°36′45″ N, 106°17′59″ E; altitude 261 m) of Chongqing Normal University, China. After being sieved (<1 mm) and air-dried, the soil contained 17.19 ± 0.62 g/kg of organic matter; 40.76 ± 2.86, 70.19 ± 2.01 and 93.84 ± 10.91 mg/kg of available nitrogen, phosphorus and potassium, respectively; and 0.78 ± 0.05, 1.02 ± 0.12 and 17.21 ± 0.38 g/kg of total nitrogen, phosphorus and potassium, respectively. When the two cotyledons started to unfold, 1 mL of RP01 or an equal volume of deionized water was inoculated into the rhizosphere.

Throughout the experimental period, cotton seedlings were randomly placed under greenhouse conditions, where the average day/night period, daytime light intensity, temperature and humidity were 12.5/11.5 h, 2000–4000 lx, 16–30 °C and 50–80%, respectively. Thirty days after inoculation, indicators, including phenotypic data, hormone levels, gene expression and microbial diversity, were measured. Three independent experiments were performed, with 30 plants per replicate.

### 4.4. Expression of Hormones, Growth and Disease Resistance-Related Genes

The levels of auxin (indole acetic acid (IAA)), brassinosteroids (BRs), salicylic acid (SA), and jasmonic acid (JA) in cotton leaves were measured using the corresponding ELISA Kits (Shanghai Preferred Biotechnology, China), with a minimum detection concentration of less than 0.1 nM and an accuracy of more than 99%. To determine the expression of genes related to growth and disease resistance, RNA was extracted from cotton roots using the MiniBEST Plant RNA Extraction Kit (TaKaRa, Maebashi, Japan). cDNA was obtained using the RT Reagent Kit for Perfect Real Time (TaKaRa). *Gh*histone3 (the gene encoding *G. hirsutum* histone 3) was used as the reference gene in quantitative reverse transcription-PCR (RT-qPCR). The expression of growth-promoting genes in the cotton root was measured in 10 µL PCR reactions containing 5 µL of SYBR Green Real-time PCR Master Mix (Bio-Rad, Hercules, CA, USA), 1 µL of root cDNA, 2 µL of ultrapure water and 1 µL each of the 10 µM forward and reverse primers (Appendix A). RT-qPCR was performed on a CFX96 instrument (Bio-Rad) using the following protocol: 94 °C for 2 min, followed by 39 cycles of 94 °C for 5 s and 60 °C for 30 s, then 95 °C for 5 s, 65 °C for 5 s and 95 °C for 5 s.

### 4.5. Assessing the Rhizosphere and Endophytic Microbiota

The cotton roots in rhizosphere soils were placed in 0.02 M of phosphate-buffered saline (pH 6.8) and incubated at 180 rpm for 20 min. After removing the roots, the suspension was centrifuged at 12,000× *g* for 10 min to collect the sediment containing the rhizosphere soil samples. The sediment was rinsed with 70% ethanol for 2 min and then with sterile water (five times). The cotton roots removed from the previous step were analyzed for endophytic microorganism content. According to the manufacturer’s instructions, total microbial genomic DNA was extracted from the cotton root and rhizosphere soil samples using the DNeasy PowerSoil Kit (Qiagen, Hilden, Germany). The DNA was stored at −20 °C for further analysis.

PCR amplification of the V3-V4 region of the bacterial 16S rRNA gene was performed using the forward primer 338F (5′-ACTCCTACGGGAGGCAGCA-3′) and the reverse primer 806R (5′-GGACTACHVGGGTWTCTAAT-3′) [59], and PCR amplification of the ITS region was performed using the forward primer ITS3F (5′-GCATCGATGAAGAACGCAGC-3′) and the reverse primer ITS4R (5′-TCCTCCGCTTATTGATATGC-3′) [60].

The PCR amplicons were purified using Agencourt AMPure Beads (Beckman Coulter, Brea, CA, USA) and quantified using the PicoGreen dsDNA Assay Kit (Invitrogen, Carlsbad, CA, USA). After individual quantification steps, equal volumes of amplicons were pooled, and paired-end 2 × 300 bp sequencing was performed on the Illumina MiSeq platform (Illumina, San Diego, CA, USA) using the MiSeq Reagent Kit v3 from Shanghai Personal Biotechnology (Shanghai, China). Data were analyzed using the free online Majorbio Cloud Platform (www.majorbio.com, accessed on 2 February 2023). Operational taxonomic units (OTUs) were clustered at a 97% similarity. Rarefaction curves [61] were generated using a reasonable amount of sequencing data (Appendix A), which resulted in a flat curve, indicating that the amount of sequencing data was large enough to reflect the vast majority of microbial diversity information in the sample. In addition, reads representing chloroplasts and mitochondria were removed prior to further analysis, as chloroplasts and mitochondria are abundant in cotton roots.

### 4.6. Whole-Genome Sequencing and Comparative Genome Analysis

The genome was sequenced by Genedenovo Biotechnology Co. (Guangzhou, China) using a PacBio RS II system (Pacific Bioscience, Menlo Park, CA, USA). To evaluate the complexity of the genome and correct the PacBio long reads, the RP01 genome was sequenced by Biozeron Co. (Shanghai, China) using the Illumina HiSeq platform (PE150 mode, Illumina, San Diego, CA, USA). RP01-coding genes were predicted using Glimmer v3.02 [62]. Then, all genes were blasted against the non-redundant (NR) (https://ftp.ncbi.nih.gov/blast/db/, accessed on 26 January 2023), Swiss-Prot (https://web.expasy.org/docs/swiss-prot_guideline.html, accessed on 26 January 2023), Pfam (http://pfam.xfam.org/, accessed on 26 January 2023), KEGG (http://www.genome.jp/kegg/, accessed on 26 January 2023), GO (http://www.geneontology.org/, accessed on 26 January 2023) and EggNOG (http://eggnog.embl.de/, accessed on 26 January 2023) databases for functional annotation using the BLASTp module. In addition, tRNAs were identified using tRNAscan-SE (v2) [63], rRNAs were identified using RNAmmer (v1.27) [64] and gene islands were predicted using IslandPath-DIMOB (v1.0.0) [65]. Comparative genomic analysis was performed by comparing the genome sequence of the RP01 strain with that of three other *B. paralicheniformis* strains.

### 4.7. Statistical Analysis

Biochemical and physiological measurements were presented as mean ± standard error (SE). Values were compared using a one-way analysis of variance (ANOVA), followed by Bonferroni’s post hoc test using the Statistical Package for the Social Sciences, v22.0 (SPSS, Chicago, IL, USA). Means among treatments were considered significantly different when the probability (*p*-value) was less than 0.05.

For the statistical analyses of 16S rRNA gene amplicon data, alpha-diversity (including the Sob, Shannon, Ace, Chao1, Simpson, coverage and others indexes) was calculated using Mothur (version v.1.30.2; https://mothur.org/wiki/calculators/, accessed on 03 February 2023). Significant differences were assessed by Student’s *t*-test. Community composition analysis (bar and pie diagrams), the Venn diagram and the heatmap were created using R version 3.3.1. The species-related network (one-way network analysis) reflected interactions between genera based on genera-to-genera correlations and was performed in Python version 2.7 using the Networkx package. Microbiome functions were predicted using PICRUSt2 from the 16S rRNA data. A non-parametric Kruskal–Wallis test was used when the data were not normally distributed. The Bray–Curtis dissimilarity metric and analysis of similarities (ANOSIM) with 999 permutations was performed when comparing groups. Those analyses were performed using the Majorbio Cloud Platform (www.majorbio.com, accessed on 13 February 2023) [65]. Unless otherwise noted, default parameters were used for all of the software.

## 5. Conclusions

The *Bacillus paralicheniformis* RP01 strain with beneficial growth-promoting activity was screened from the root surface of *Brassica chinensis* using a phosphate-solubilizing medium. RP01 inoculation significantly increased plant root length and BR content and upregulated the expression of growth-related genes. Simultaneously, it increased the number of beneficial bacteria that promoted plant growth and reduced the number of detrimental bacteria. The genome annotation findings also revealed that RP01 possesses a variety of growth-promoting mechanisms and a tremendous growth-promoting potential. This study isolated a highly potential PGPB and elucidated its possible direct and indirect growth-promoting mechanisms. Moreover, RP01 can also produce many secondary metabolites with unknown functions, which may play a role in agricultural and pharmaceutical applications. Our study results will help enrich the PGPB library and provide a reference for plant–microbe interactions.

## Figures and Tables

**Figure 1 ijms-24-07227-f001:**
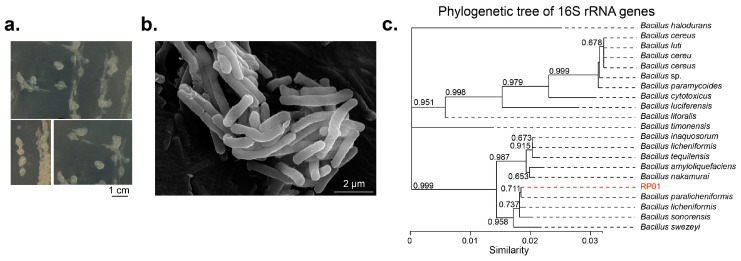
Morphological characteristics of *Bacillus paralicheniformis* RP01. (**a**) Colony phenotype of *B. paralicheniformis* RP01 on LB medium, scale bar represents 1 cm. (**b**) Cellular morphology under scanning electron microscope after culture for 16 h at 37 °C, scale bar represents 2 μm. (**c**) Molecular evolutionary tree for RP01 and other 20 strains in GenBank database. The tree was built using the 16S rRNA region gene (27F and 1492R) by the neighbor joining method with a bootstrap value of 1000.

**Figure 2 ijms-24-07227-f002:**
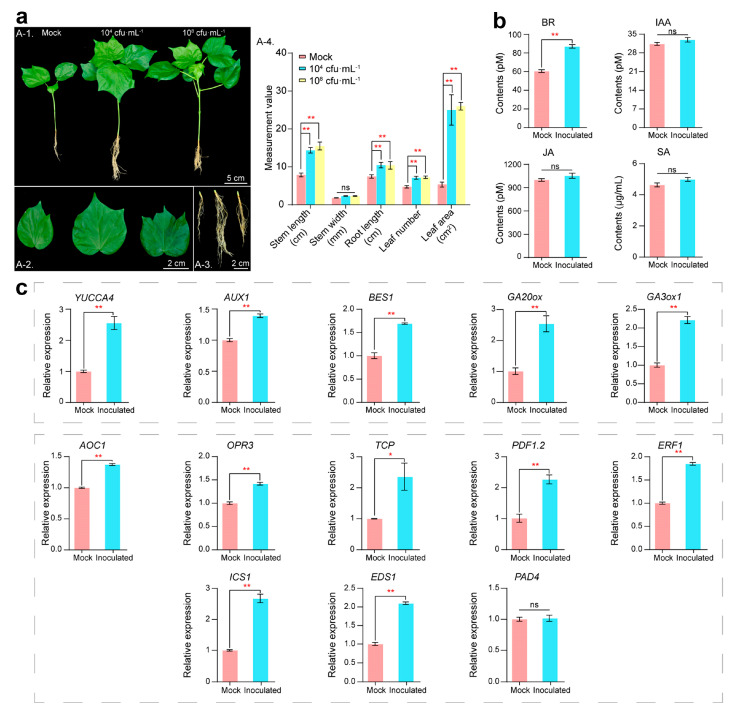
Phenotypic changes in cotton 30 days after inoculation with *B. paralicheniformis* RP01. (**a**) The changes in phenotype of the whole plant (A-1), enlarged leaf (A-2) and enlarged root (A-3). Plant biomass data (A-4). Leaf area is measured as the area of the third leaf from the top bud. Treatment groups from left to right are the mock, 10^4^ cfu/mL and 10^8^ cfu/mL groups. (**b**) Hormone content of cotton leaves. (**c**) Hormone-related genes by RT-qPCR. Error bars represent standard error (SE). Statistical significance was calculated using one-way analysis of variance (ANOVA) followed by Bonferroni’s post hoc test. The asterisk (*) indicates a significant difference at *p* < 0.05; (**) indicates *p* < 0.01.

**Figure 3 ijms-24-07227-f003:**
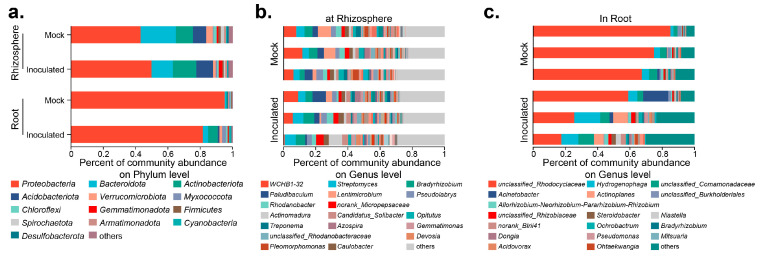
Microbial composition in rhizosphere and roots 30 days after *B. paralicheniformis* RP01 inoculation. (**a**) Relative abundance at phylum level in different groups. (**b**) Relative abundance at genus level in the rhizosphere samples. (**c**) Relative abundance at genus level in the root samples.

**Figure 4 ijms-24-07227-f004:**
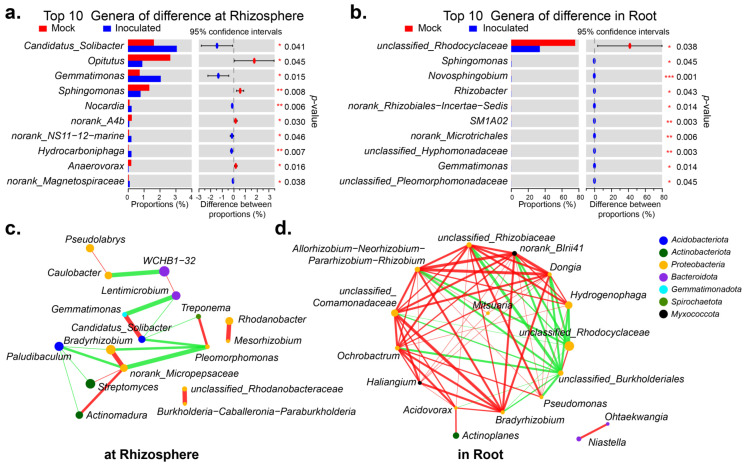
Important genera in rhizosphere and roots. The difference of top 10 genera when inoculated with *B. paralicheniformis* RP01 and without in rhizosphere (**a**) and in roots (**b**). Sorted by abundance. The species-related network reflected interactions between genera based on abundance, showing top 20 genera-to-genera correlations in rhizosphere (**c**) and in root. (**d**). The size of the nodes indicates the abundance of species, and different colors represent different phyla. The color of the connection line indicates positive and negative correlation, where red indicates positive correlation, and green indicates negative correlation. The thickness of the line indicates the size of the correlation coefficient; the thicker the line, the higher the correlation between species. The more lines, the closer the connection between that species and other species. *, *p* < 0.05; **, *p* < 0.01; ***, *p* < 0.001.

**Figure 5 ijms-24-07227-f005:**
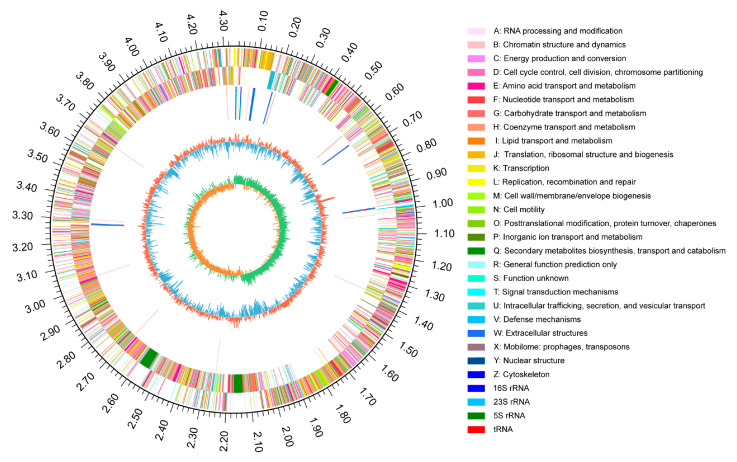
Schematic of the complete *B. paralicheniformis* RP01 genome. Rings represent the following features labeled from the outside to the center: The outermost circle represents the scale in bp. The second ring represents positive strand genes and the third ring represents negative strand genes. Each color patch represents a COG functional classification. The fourth ring represents rRNA and tRNA. The fifth ring represents the GC content (red indicates GC content above the mean, blue indicates GC content below the mean). The innermost ring shows the GC skew (GC skew = (G − C)/(G + C); green means greater than 0, orange means less than 0).

**Figure 6 ijms-24-07227-f006:**
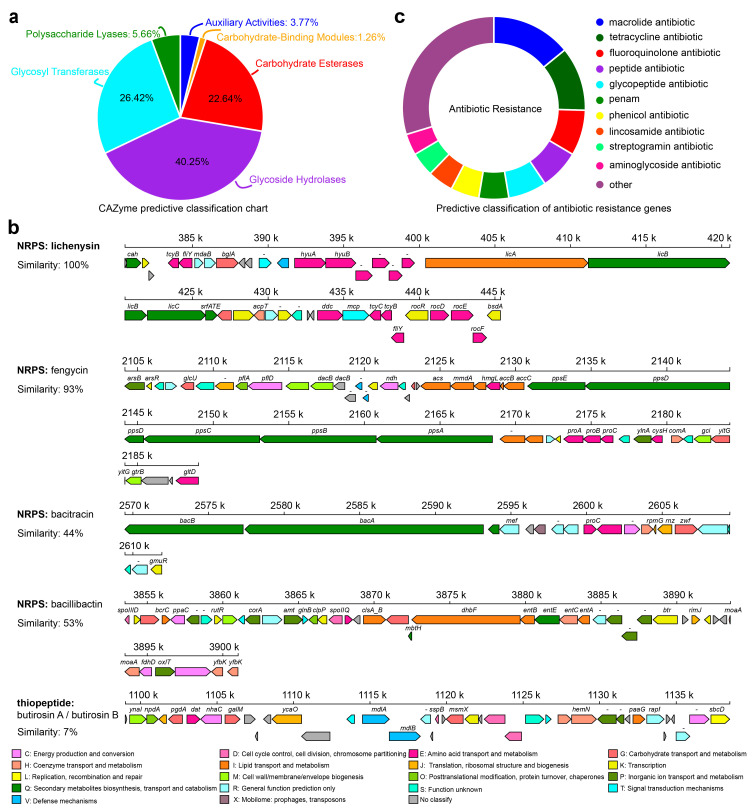
Metabolic system analysis of *B. paralicheniformis* RP01. (**a**) Carbohydrate-active enzyme prediction in RP01. (**b**) The secondary metabolite gene cluster prediction of RP01, and the colors of boxes based on the COG classification. (**c**) Drug resistance gene prediction in RP01 genome.

**Figure 7 ijms-24-07227-f007:**
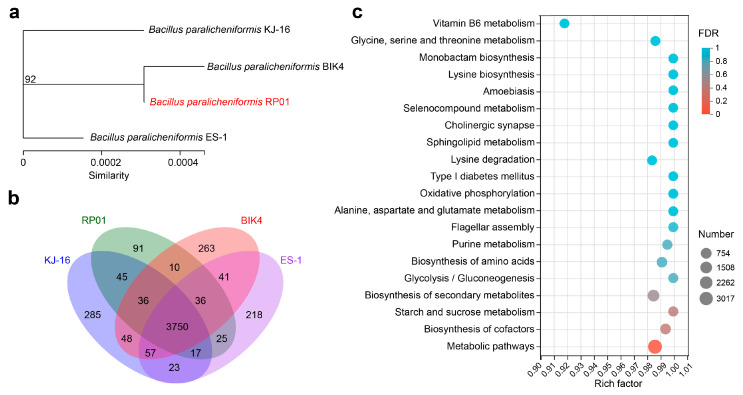
Comparative analysis of the *B. paralicheniformis* RP01 genome. (**a**) Molecular evolutionary tree for the four strains. The tree was built using housekeeping genes (*dnaG*, *frr*, *infC*, *nusA*, *pgk*, *pyrG*, *rplA*, *rplB*, *rplC*, *rplD*, *rplE*, *rplF*, *rplK*, *rplL*, *rplM*, *rplN*, *rplP*, *rplS*, *rplT*, *rpmA*, *rpoB*, *rpsB*, *rpsC*, *rpsE*, *rpsI*, *rpsJ*, *rpsK*, *rpsM*, *rpsS*, *smpB* and *tsf*) by maximum likelihood method with a bootstrap value of 1000. (**b**) Venn diagram shows the number of genes in all four strains. (**c**) KEGG enrichment analysis of homologous genes in four strains. The X-axis represents rich rate (refers to the ratio of the number of genes enriched in the pathway to the number of annotated genes; a larger ratio indicates greater enrichment). The size of the point indicates the number of genes in this pathway and the color of the point indicates the significance of enrichment.

**Table 1 ijms-24-07227-t001:** General information on Bacillus paralicheniformis RP01 and other three *B. paralicheniformis* strains.

Strain	*B. paralicheniformis* RP01	*B. paralicheniformis* KJ-16	*B. paralicheniformis* BIK4	*B. paralicheniformis* ES-1
Genome size (bp)	4,338,611	4,520,660	4,422,539	4,397,844
GC content	45.95%	45.76%	45.48%	45.75%
Gene No.	4222	4540	4511	4378
tRNA No.	81	77	80	81
5S rRNA No.	8	6	3	8
16S rRNA No.	8	2	1	1
23S rRNA No.	8	1	1	1
G + C%	45.9546	45.4788	45.7614	45.7472
GI No.	8	14	8	7
Prophage No.	1	5	2	3
CAZyme	159	165	160	160
Secondary metabolite	12	13	14	12
Antibiotic resistance	268	270	269	273
Reference	This paper	[36]	[35]	[37]
accession No.	CP118744	LBMN01	GCA_019336205.1	CP083398
Characteristic	Rhizobacteria; promote plant growth	Fermented soybean product	Rhizobacteria; promote plant growth	Salt mine sodic soil; broad-spectrum antimicrobial and halotolerant

## Data Availability

Not applicable.

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
