# Peer review of "Bacillus paralicheniformis RP01 Enhances the Expression of Growth-Related Genes in Cotton and Promotes Plant Growth by Altering Microbiota inside and outside the Root"

_ijms, 2023, doi:10.3390/ijms24087227_

Round 1

Reviewer 1 Report

Dear Author, 

the paper "Bacillus paralicheniformis RP01 enhances the expression of growth-related genes in cotton and promotes plant growth by altering microbiota inside and outside the root" in my opinion is very interest and well write. in my opinion no additional experiment needed. 

Only a minor revision in the test are request: 

in the introduction control the parenthesis of reference because are double in the end. 

Control in the text there name of bacteria in particular Candidatus_Solibacter, in some sentence the symbol _ is missing; and control the Italic character between phylum, family and genus

Line 102 and 111-119: remove double ] in the reference

Line 122-123 and 129: the sentence is not well write

Line 356: add the name of the medium and not only the acronym

Line 361: add comma between phenotypical and physiological 

Author Response

Comments:
The paper "Bacillus paralicheniformis RP01 enhances the expression of growth-related genes in cotton and promotes plant growth by altering microbiota inside and outside the root" in my opinion is very interest and well write. in my opinion no additional experiment needed.

Response: Thank you very much for your favorable consideration.

  1. - In the introduction control the parenthesis of reference because are double in the end.

Response: Many thanks for pointing this out. We are sorry for our carelessness and have revised.

  1. - Control in the text there name of bacteria in particular Candidatus_Solibacter, in some sentence the symbol _ is missing; and control the Italic character between phylum, family and genus.

Response: Thank you very much for your suggestion. We have added “_” in the revised manuscript and Figures, such as “Candidatus_Solibacter” and “unclassified_Comamonadaceae”.

According to the ICPN (International Code of Nomenclature of Prokaryotes, https://doi.org/10.1099/ijsem.0.000778), all the categories of Prokaryotes need to be italic including genus and other higher categories. We checked the full text based on this.

  1. - Line 102 and 111-119: remove double ] in the reference

Response: Many thanks for pointing this out. We are sorry for our carelessness and have revised.

  1. - Line 122-123 and 129: the sentence is not well write

Response: Thank you very much for your suggestion. We have revised.

  1. - Line 356: add the name of the medium and not only the acronym

Response: Thank you very much for your suggestion. We have added.

  1. - Line 361: add comma between phenotypical and physiological

Response: Many thanks for pointing this out. We have fixed the error.

Reviewer 2 Report

I have carefully reviewed the manuscript entitled "Bacillus paralicheniformis RP01 enhances the expression of growth-related genes in cotton and promotes plant growth by altering microbiota inside and outside the root" by Xu et al. submitted to Journal International Journal of Molecular Sciences. This study investigated the growth-promoting mechanism of Bacillus paralicheniformis RP01 using cotton as the host to increase crop yield and improve the soil conditions for sustainable agriculture. The manuscript described the possible direct and indirect growth-promoting mechanisms of RP01 by testing cotton phenotype, physiological characteristics, genome and microbiome. Overall, this manuscript reflects a good study and provides a lot of useful data. These data can advance the understanding involved in growth-promoting mechanisms by PGPRs and the interactions of PGPRs and their hosts. In this manuscript there are some minor issues that need to be improved.

1. In fact, Bacillus paralicheniformis RP01 was isolated from the root surface of Brassica chinensis. However, in this study, why was cotton selected as a host?

2. Bacillus paralicheniformis RP01 is a kind of phosphate-solubilizing microorganisms. The data including soil available P content and cotton P uptake should be added into manuscript.

3. In pot experiment, two treatments were performed (104 and 108 cfu/ml), however, the authors did not describe clearly the samples in which treatment was employed for subsequent cotton physiological and microbiome analyses.

4. The introduction section is relatively simple. Some contents involved in the mechanisms of PGPRs should be supplemented.

5. The data associated with alpha diversity of microbiome should be supplemented since alpha diversity is an important microbiome attribute impacting crop growth and development.

6. Line 424-428, bacterial primer pair should be 338F/806R, and the name of fungal primer pair should be added.

Author Response

Comments:
I have carefully reviewed the manuscript entitled "Bacillus paralicheniformis RP01 enhances the expression of growth-related genes in cotton and promotes plant growth by altering microbiota inside and outside the root" by Xu et al. submitted to Journal International Journal of Molecular Sciences. This study investigated the growth-promoting mechanism of Bacillus paralicheniformis RP01 using cotton as the host to increase crop yield and improve the soil conditions for sustainable agriculture. The manuscript described the possible direct and indirect growth-promoting mechanisms of RP01 by testing cotton phenotype, physiological characteristics, genome and microbiome. Overall, this manuscript reflects a good study and provides a lot of useful data. These data can advance the understanding involved in growth-promoting mechanisms by PGPRs and the interactions of PGPRs and their hosts. In this manuscript there are some minor issues that need to be improved.

Response: Thank you very much for your favorable consideration.

  1. - In fact, Bacillus paralicheniformisRP01 was isolated from the root surface of Brassica chinensis. However, in this study, why was cotton selected as a host?

Response: Many thanks for your comments. We have conducted experiments in several host, including pakchoi cabbage (Brassica chinensis), tobacco (Nicotiana tobacum K326) and cotton (Gossypium hirsutumYumian-1) with good plant-promoting results (Figure 2a and Figure S1). The results of the previous experiments showed the best performance in cotton, so we chose cotton as host for our preliminary investigation.

  1. - Bacillus paralicheniformisRP01 is a kind of phosphate-solubilizing microorganisms. The data including soil available P content and cotton P uptake should be added into manuscript.

Response: Thank you very much for your suggestion. We tested the phosphate-solubilizing ability of RP01 (Table S1.), and in this manuscript we showed a macroscopic view of the mechanism of RP01 in plant growth promotion from several aspects.

Subsequently, we will make a more in-depth special report on the role of phosphate-solubilizing mechanism of RP01 in plant growth promotion, which I will show in the next article to ensure the integrity of the data.

  1. -In pot experiment, two treatments were performed (104and 108cfu/ml), however, the authors did not describe clearly the samples in which treatment was employed for subsequent cotton physiological and microbiome analyses.

Response: Thank you very much for your suggestion.

For the growth-promoting effect was the same for both low and high inoculations (Figure 2a), we provided an explanation in Line 89-91: “Therefore, we compared cotton seedlings inoculated with low concentrations (104 cfu/mL, inoculation group) of RP01 and sterile water control (mock group) to analyze how they promote plant growth.”

  1. - The introduction section is relatively simple. Some contents involved in the mechanisms of PGPRs should be supplemented.

Response: Thank you very much for your suggestion. We have added in the revised manuscript.

  1. - The data associated with alpha diversity of microbiome should be supplemented since alpha diversity is an important microbiome attribute impacting crop growth and development.

Response: Many thanks for pointing this out. We analyzed the alpha diversity showing no significant differences, so it is not mentioned in the text, but all data were shown in the original Table S4.

Your considerations are very valuable and we have added the relevant descriptions in the revised manuscript (now in Table S3).

  1. - Line 424-428, bacterial primer pair should be 338F/806R, and the name of fungal primer pair should be added.

Response: Many thanks for pointing this out. We have revised.
